# The Upstream 1350~1250 Nucleotide Sequences of the Human *ENDOU-1* Gene Contain Critical *Cis*-Elements Responsible for Upregulating Its Transcription during ER Stress

**DOI:** 10.3390/ijms242417393

**Published:** 2023-12-12

**Authors:** Hung-Chieh Lee, Hsuan-Te Chao, Selina Yi-Hsuan Lee, Cheng-Yung Lin, Huai-Jen Tsai

**Affiliations:** 1Department of Life Science, Fu-Jen Catholic University, New Taipei City 242062, Taiwan; 2Faculty of Sciences and Engineering, Maastricht University, 6211 LK Maastricht, The Netherlands; 3Institute of Biomedical Sciences, Mackay Medical College, New Taipei City 25245, Taiwan

**Keywords:** ER stress, *ENDOU-1* gene, promoter analysis, *cis*-acting element, transcriptional regulation, zebrafish, transcription factor, HEK-293T

## Abstract

*ENDOU-1* encodes an endoribonuclease that overcomes the inhibitory upstream open reading frame (uORF)-trap at 5′-untranslated region (UTR) of the *CHOP* transcript, allowing the downstream coding sequence of *CHOP* be translated during endoplasmic reticulum (ER) stress. However, transcriptional control of *ENDOU-1* remains enigmatic. To address this, we cloned an upstream 2.1 kb (−2055~+77 bp) of human *ENDOU-1* (pE2.1p) fused with reporter luciferase (luc) cDNA. The promoter strength driven by pE2.1p was significantly upregulated in both pE2.1p-transfected cells and pE2.1p-injected zebrafish embryos treated with stress inducers. Comparing the luc activities driven by pE2.1p and −1125~+77 (pE1.2p) segments, we revealed that *cis*-elements located at the −2055~−1125 segment might play a critical role in *ENDOU-1* upregulation during ER stress. Since bioinformatics analysis predicted many *cis*-elements clustered at the −1850~−1250, we further deconstructed this segment to generate pE2.1p-based derivatives lacking −1850~−1750, −1749~−1650, −1649~−1486, −1485~−1350 or −1350~−1250 segments. Quantification of promoter activities driven by these five internal deletion plasmids suggested a repressor binding element within the −1649~−1486 and an activator binding element within the −1350~−1250. Since luc activities driven by the −1649~−1486 were not significantly different between normal and stress conditions, we herein propose that the stress-inducible activator bound at the −1350~−1250 segment makes a major contribution to the increased expression of human *ENDOU-1* upon ER stresses.

## 1. Introduction

The endoplasmic reticulum (ER) is a critical organelle involved in the synthesis and folding of proteins in order to maintain normal cell function and homeostasis [1,2,3]. The Integrated Stress Response (ISR) is a cellular stress response that aims to restore cellular homeostasis upon different types of extrinsic or intrinsic stresses. The ISR can be stimulated by different environmental and pathological conditions, including nutrient deprivation, oxidative stress, protein homeostasis defects, and viral infection [4,5,6,7,8,9,10,11]. The ISR can also be stimulated by intrinsic ER stress, which is caused by the accumulation of unfolded or misfolded proteins within the ER [12]. The ISR is primarily a pro-survival homeostatic program, aiming to optimize adaptive cellular response to stress. The ISR restores cellular balance by reprogramming gene expressions in response to different environmental and pathological conditions, including oxidative stress, proteostasis, viral infection, and nutrient deprivation [13,14]. However, once the stress cannot be removed, the ISR drives signaling toward cell death, fro example, through apoptosis-inducing factors which will act as a prodeath effector to trigger DNA cleavage and parthanatos [15]. For example, dysregulated ISR signaling can be involved in cognitive disturbances, neurodegenerative diseases, tumor malignancies, diabetes, and metabolic imbalances [16].

The Eukaryotic ENODU ribonuclease family consists of several enzymes that share amino acid homology with XendoU [17,18]. This ENDOU family is broadly conserved from viruses to humans. For example, Nsp15, a viral orthologue of XendoU, plays an essential role in coronavirus replication [19] and host immune suppression [20]. *Xenopus* XendoU is a Uridylate-specific, divalent cation-dependent enzyme that produces molecules with 2′,3′-cyclic phosphate ends, a unique characteristic of this particular class of RNases involved in generating U16 and U86 small nucleolar RNAs (snoRNAs) through the cleavage of pre-mRNAs encoded within the third intron of the L1 ribosomal protein gene [17,18,21,22,23]. Human PP11 has been recently categorized as a member of the XendoU family, i.e., ENDOU. Despite its annotated function as a putative serine protease, PP11/ENDOU is a placental-specific endoribonuclease. Moreover, the dysregulated expression of PP11/ENDOU in tumor tissues may be associated with carcinogenesis [24]. Nevertheless, *Xenopus* XendoU and human ENDOU-2 play similar roles in cellular processes, including the regulation of ER structure, RNA degradation, and cell survival [25,26]. Moreover, *C. elegans* ENDOU-2 was reported to be involved in regulating cold tolerance and synaptic remodeling through caspase signaling [27]. Mammalian *ENDOU-1* mRNAs are rapidly increased in the presence of stresses, such as heat shock and hypoxia, subsequently inducing eIF2α phosphorylation, suggesting that ENDOU-1 protein may play a role in stress response and serve as a novel stress factor [28]. However, control over the transcription of *ENDOU-1* gene in response to ER stresses remains to be elucidated.

In general, the eukaryotic promoter can roughly be divided into two parts: proximal and distal. The proximal region of eukaryotic promoter is believed to be responsible for correctly assembling the RNA polymerase II complex at the right position, and for directing a basal level of transcription. The distal region is believed to contain *cis*-elements that regulate spatiotemporal expression. It is well known that 98% of DNA sequences are non-protein-coding regions in the human genome [29]. These non-coding regions contain a variety of *cis*-regulatory elements which interact with transcription factors to orchestrate the expression of specific genes spatiotemporally. Therefore, identification of *cis*-acting and *trans*-acting elements is important to gain insight into the regulatory mechanisms controlling *ENDOU-1* expression

In this study, we first cloned an upstream 2.1 kb (−2055~+77 bp) of human *ENDOU-1* fused with luciferase (luc) reporter. Next, we compared the quantification of luc expression activities driven by different deleted upstream segments during normal and stress condition. Evidence from both in vivo and in vitro systems suggests two indispensable *cis*-acting elements within the −1850~−1250 segment. These *cis*-acting elements are involved in controlling the transcription of −2.1-kb human *ENDOU-1* during normal and stress conditions. One is a repressor binding motif located within the −1649~−1486 region, but the other is an activator binding motif located within the −1250~−1350 region of *ENDOU-1*. Since the −1649~−1486 segment was independent of upregulated *ENDOU-1* at stress, while gene expression driven by the −1250~−1350 segment could be significantly enhanced after ER stress, we herein propose that the stress-inducible *cis*-element bound by an activator within the −1350~−1250 segment makes a major contribution to the increased expression of *ENDOU-1* upon ER stress treatment.

## 2. Results and Discussion

### 2.1. Cloning the Upstream DNA Fragment of the Human ENDOU-1 Gene in Which the Stress-Responsive Element Was Inclusive

To understand whether the upregulation of human *ENDOU-1* in response to ER stress is controlled at transcriptional level, we aimed to clone the upstream regulatory region of human *ENDOU-1* and determine if this regulatory region could be activated during ER stress. First, we cloned an upstream 2.1 kb (−2.1 kb) fragment of human *ENDOU-1* and subcloned it into the expression vector containing firefly-luc reporter (Flu) to generate pE2.1p plasmid. The dual-luc activity was performed in HEK-293T cells, in which pE2.1p (100 ng) was co-transfected with phRG-TK (20 ng, containing Renilla-luc reporter (Rlu), driven by thymidine kinase (TK) promoter which served as an internal control) for 24 h and treated with either DMSO (control group) or Thapsigargin (TH; stress group) for 2 or 6 h, followed by analysis of luc activity. Compared to DMSO-treated control cells, the luc activity driven by pE2.1p was increased 2.1-fold in cells treated with TH for 2 h (Figure 1A). Moreover, the stress-induced effect on the strength of *ENDOU-1* promoter positively correlated with the duration of stress treatment. For example, luc activity was increased 3.1-fold after TH treatment for 6 h (Figure 1A).

Next, we employed zebrafish embryos to perform an in vivo study. Zebrafish embryos at the one-cell stage were simultaneously microinjected with a mixture of pE2.1p and phRG-TK and treated with heat shock at 72 h post-fertilization (hpf). We found that luc activity increased ~2.07-fold compared to that of embryos kept at a normal condition (28.5 °C) during 96 hpf (Figure 1B). Based on consistent results obtained from both in vitro and in vivo systems, we concluded that an upstream 2.1 kb segment of human *ENDOU-1* is suitable for conducting a detailed investigation of *cis*-elements potentially involved in the transcription of human *ENDOU-1* after the induction of stress.

### 2.2. Reporter Gene Expression Driven by the Upstream 2.1-kb of Human ENDOU-1 Could Be Upregulated When the Stress Inducers Were Applied to Cells

We examined whether the upstream 2.1 kb fragment of *ENDOU-1* was responsive to various stress-inducing drugs, as described below. Dual-luc activity was assessed in HEK-293T cells, in which pE2.1p (100 ng) was co-transfected with phRG-TK (20 ng) for 24 h and treated with DMSO (control group), TH, Anisomycin, CoCl_2_ or Poly I:C for 4 h, followed by analysis of luc activity.

TH is a non-competitive inhibitor of the sarco/endoplasmic reticulum Ca^2+^ ATPase (SERCA). Depletion of ER calcium stores leads to ER stress and activation of the unfolded protein response (UPR). Anisomycin is a potent protein synthesis inhibitor, which interferes with protein and DNA synthesis by inhibiting peptidyl transferase or the 80S ribosome system, resulting in ribotoxic stress. CoCl_2_ treatment mimics hypoxic stress, since CoCl_2_ inhibits prolyl hydroxylase domain (PHD) enzymes (oxygen sensors) through replacement of Fe^2+^ with Co^2+^, making these enzymes unable to hydroxylate hypoxia-inducible factor-1 α (HIF-1α), thus activating HIF signaling, resulting in the induction of a hypoxia-like condition [30]. Poly I:C is a synthetic double-stranded RNA that is used experimentally to model viral infections in vivo. Copper (Cu), a transition metal, is an essential trace element in human and animal nutrition at low concentration. However, Cu^2+^ at high concentration can induce ER stress and promote apoptosis, since it activates the CHOP, JNK and Caspase-12 signaling pathways [31].

As shown in Figure 2, we found that HEK-293T cells treated with all stress-inducing drugs could increase promoter activity driven by pE2.1p (Figure 2A). Western blot analysis demonstrated that the expression levels of CHOP and p-eIF2α, two important markers for ER stress [28], were induced significantly (Figure 2B), indicating that the stress-inducing drugs used in this study caused the response of HEK-293T cells to stress. Interestingly, we noticed that Poly I:C could strongly induce *ENDOU-1* promoter activity, whereas Rapamycin could only slightly induce it (Figure 2A), suggesting that promoter strength driven by the −2055~+77 DNA fragment of the human *ENDOU-1* gene is dependent on the stress inducer applied to cells, in this case HEK-293T.

### 2.3. Computational Analysis Proposes That the Distal Region of the −2.1 kb ENDOU-1 Fragment May Contain Critical Regulatory Elements

Next, we wanted to identify *cis*-regulatory responsive elements (CREs) located at the −2250~+77 DNA fragment of human *ENDOU-1* that might play a role in upregulating *ENDOU-1* transcripts during ER stress. To accomplish this, we employed *Tfsitescan* and PROMO to analyze potential transcription factor binding sites (TFBSs). Owing to the limited number of base pairs per scan, we divided the −2.1 kb fragments into two parts with an overlap of 90 bp. Thus, the −2055~−975 fragment was scanned first, followed by scanning the −1065~00 fragment. The results revealed that the distal region of the −2.1 kb fragment (the −2055~−975 fragment) contains more potential TFBSs of relative significance than those contained in the proximal region (the −1065~00 fragment) (Appendix A). We further asked if important regulatory elements were harbored within the −2055~−975 region. Results showed that this region contained five putative transcription factors (Appendix A), including nuclear factor Y (NF-Y), Yin Yang 1 (YY1), activator protein-1 (AP-1), c-Erythroblast transformation specific-1 (c-Ets-1), and nuclear factor-κB (NF-κB), all previously reported as critical CREs in the promoter regions of other ER stress-induced genes [32,33,34,35,36,37,38,39]. Therefore, we hypothesized that regulatory elements contained in the −2055~−975 region may be critical for controlling the upregulation of *ENDOU-1* in response to ER stress.

### 2.4. The −2055~−1125 Segment of Human ENDOU-1 Contains Cis-Acting Elements Involved in Increasing Promoter Activity under Stress Conditions

Focusing on the distal region of the −2.1 kb fragment of *ENDOU-1*, we continued to seek to identify *cis*-regulatory elements capable of upregulating the expression of *ENDOU-1* during stress. To accomplish this, we constructed serial deletion clones from the upstream fragment of *ENDOU-1* fused with luc reporter cDNA. Based on the restriction enzyme map, we found a XhoI-cutting site at the −1125 nucleotide (nt) position of *ENDOU-1*. Taking advantage of this cutting site, XhoI was used to cut the −2250~+77 DNA fragment to obtain a resultant −1125~+77 fragment. This shortened DNA fragment was then inserted into the pGL3-basic vector to generate a plasmid pE1.2p containing this −1125~+77 fragment, allowing us to determine whether the increased *ENDOU-1* expression during ER stress is independent of the −1125~+77 segment. In order to compare the luc activities driven by the −2055~+77 (pE2.1p) and −1125~+77 (pE1.2p) segments, we first co-transfected HEK-293T cells with pE1.2p (containing a −1125~+77 segment) and phRG-TK and then treated with TH. The results exhibited a ~0.77-fold decrease compared to that of cells treated with DMSO control (Figure 3A), suggesting that the luc activities did not display much difference among pE1.2p-injected cells, TH-treated cells, and untreated cells. However, the luc activity of TH-treated HEK-293T co-transfected with pE2.1p (containing a −2055~+77 segment) and phRG-TK exhibited a ~2.36-fold increase (Figure 3A), suggesting that the *cis*-elements located at −2055~−1125 region might play a critical role in *ENDOU-1* upregulation during ER stress.

To further confirm the above in vitro finding, the dual-luc assay was also performed in zebrafish embryos to explore any difference between luc activities driven by pE2.1p and pE1.2p segments in vivo. The results demonstrated that luc activities did not display much difference (around 0.56-fold decrease) among pE1.2p-injected embryos, TH-treated embryos and untreated control embryos (Figure 3B). However, pE2.1p-injected zebrafish embryos treated with heat shock displayed ~2.29-fold increase in luc activity compared to that of embryos without TH treatment (Figure 3B). The consistency between in vitro and in vivo evidence strongly suggests that the upstream fragment of human *ENDOU-1* that enables an increase in promoter activity under stress condition is independent of the −1125~+77 region. In other words, the *cis*-acting elements contained in the −2055~−1125 segment are responsible for the response of *ENDOU-1* to ER stress.

### 2.5. Two Indispensable Cis-Acting Elements within the −1850~−1250 Segment Controlled the Expression of −2.1-kb Human ENDOU-1 under Normal and Stress Conditions

The above results demonstrated that the −2055~−1125 fragment may contain *cis*-elements responsible for controlling the transcription of *ENDOU-1* during stress. Nevertheless, we employed Tfsitescan and PROMO to analyze potential TFBSs, and based on this analysis, many TFs predicted to target the *ENDOU-1* promoter region were clustered at the −1850~−1250 region. Therefore, to further identify *cis*-regulatory elements involved in controlling the transcription of *ENDOU-1*, we employed a PCR-based internal deletion strategy to deconstruct the −1850~−1250 region of pE2.1p in more detail. To accomplish this, we constructed five internal deletion clones, including −1850~−1750, −1749~−1650, −1649~−1486, −1485~−1350 and −1350~−1250, and their promoter activities were analyzed, both in vitro and in vivo. According to our results, reporter activity driven by the −1649~−1486-deleted-pE2.1p exhibited a 4.11-fold increase over that of pE2.1p during normal condition (Figure 4A). However, unexpectedly, the reporter activity driven by this same deleted segment still exhibited about a 4-fold increase over that of pE2.1p under stress conditions (Figure 4A). Unlike pE2.1p and its deleted derivatives, including −1850~−1750, −1749~−1650 and −1485~−1350, which could significantly upregulate luc activity under stress, this −1649~−1486-deleted-pE2.1p fragment failed to induce reporter activity at higher levels under stress conditions.

Moreover, the results obtained from the above in vitro system were consistent with those obtained from our in vivo system. For example, when this −1649~−1486-deleted-pE2.1p fragment was injected into zebrafish embryos, the luc activity displayed a 5.9-fold increase over that of pE2.1p-injected embryos under normal conditions (Figure 4B). Again, the luc activity of the −1649~−1486-deleted-pE2.1p-injected embryos was maintained around a 5.4-fold increase, even after the induction of stress (Figure 4B). Consistent with the in vitro study, this −1649~−1486-deleted-pE2.1p was independent of further inducing luc expression to reach a higher level under stress. Therefore, to explain the failure of the −1649~−1486-deleted-pE2.1p segment to further induce luc expression and reach a higher level under stress conditions, as confirmed in both in vitro and in vivo studies, we looked to the repressor binding motif. Thus, we speculate that repressor bound at −1649~−1486 may block transcription under normal conditions, but exit the element exposed under stress conditions, resulting in the −1649~−1486-deleted-pE2.1p being unable to induce luc expression to reach a higher level.

To better understand a phenomenon whereby the repressor binds to an element under normal conditions, but exits during stress, we cite a convenient analogy. Inhibitors of DNA-binding (ID) proteins are a class of helix–loop–helix (HLH) transcription factors that serve as repressors to act as dominant negative antagonists of other basic HLH proteins through the formation of non-functional heterodimers [40]. During ER stress, the activation of Inositol requires enzyme 1α (IRE1α) to reduce ID1 level through degradation of Id1 mRNA [41]. Id-1 is thus downregulated in hypoxic cells via transcriptional repression mediated by ATF-3, and the effects would be expected to promote cell growth arrest [42]. To more explicitly make the comparison, we speculate that the repressor bound at −1649~−1486 may block transcription under normal condition, but exit the element exposed during stress conditions, resulting in −1649~−1486-deleted-pE2.1p being able to induce luc expression to reach a higher level at under normal conditions (while unable to induce expression under stress conditions). The case of HLH transcription factors is analogous, because transcriptional repressor Id-1 is expressed under normal conditions, but reduced under stress conditions.

Additionally, it is theoretically possible that the proposed repressor might have dual functions. That is, it might repress transcription under normal conditions, but induce transcription under stress conditions by the recruitment of as-yet unknown cofactors. To clarify this hypothetical, we cite the activation of NF-κB to maintain myogenic cells in an undifferentiated state, whereas it acts as a repressor during myocyte differentiation [43,44]. In another example, the role of Myc protein in the transcriptional activation and repression of target genes may depend on the unique factors it recruits, such as the constellation of other transcription factors proximal to the binding site, or even the chromatin context of the target gene [45,46,47].

Meanwhile, we noticed that the −1350~−1250 segment was also important for controlling *ENDOU-1* promoter transcription, since we found that luc activity driven by −1350~−1250-deleted-pE2.1p was substantially reduced both under normal and stress conditions (Figure 4A). Similar results were also obtained from in vivo experiment. For example, Zebrafish embryos injected with −1350~−1250-deleted-pE2.1p showed a significant decrease in luc activity, exhibiting a 0.52 and 0.53-fold decrease compared to pE2.1p-injected embryos under normal and stressed conditions, respectively (Figure 4B). This evidence suggested that the −1350~−1250 segment might contain an enhancer or activator-binding motif which facilitates the upregulated transcription of human *ENDOU-1* promoter.

### 2.6. TFBSs Predicted in the −1650~−1485 and −1350~−1250 Segments

Using Tfsitescan and PROMO software, some putative repressor-binding sites within −1650~−1485 segment were predicted. For example, Appendix A showed (1) growth factor independence 1 (GFI-1), a zinc finger transcription factor that mediates transcriptional repression, mainly by recruiting histone-modifying enzymes to its target genes [48]; (2) Ets-2 repressor factor (ERF), which represses extrasynaptic utrophin-A promoter activity in muscle [49]; and (3) C/EBP-ϵ27 and C/EBP-ϵ14 repressors, which negatively regulate the eosinophil-specific P2 promoter of the MBP gene [50,51]. However, it should be noted that Ets-2 and C/EBP could act as either activators or repressors, depending on the associated binding factors they recruit [50,51].

Some putative activator-binding sites within the −1350~−1250 segment were also predicted (Appendix A). For example, we found binding sites for (1) YY1, which has been reported to increase the promoter activity of the GRP78/BiP gene, a prosurvival ER chaperone, during ER stress [32]; and (2) transcription factor c-Est-1, which can be upregulated by the IRE1α/XBP1 signaling. The increased c-Ets-1 can activate the Mcl-1 promoter in human melanoma cells against apoptosis upon ER stress [34]. We also found binding sites for (3) the activator protein-1 (AP-1), which is up-regulated under ER stress conditions. Many studies reported that AP-1 is required for CHOP induction through binding to the *CHOP* promoter in response to various stresses [36,52,53]. Moreover, Klymenko et al. [37] identified AP-1 (−246 to −240) and c-Ets-1 (−205 to −199) binding sites within the *CHOP* promoter, which are necessary for ER-stress-induced *CHOP* gene activation. Yet, it is important to note that when using bioinformatic tools to analyze the *cis*-regulatory elements within a promoter, any potential TFBSs may be shown. However, due to the short and variable nature of TFBSs, it is highly plausible that false-positive elements are included [54]. Thus, after the number of putative *cis*-acting elements for studying is narrowed down through bioinformatic analysis, we should be able to firmly confirm that these predicted *cis*-acting elements do indeed have the ability to cause upregulation of reporter gene expression; an additional promoter activity assay must therefore be performed.

In this study, based on bioinformatics tools and deletion analysis of promoter activity, we conclude that the −1350~−1250 DNA fragment contains an activator-binding motif which is critical in upregulating the transcription of the human *ENDOU-1* promoter under stress condition. AP-1 and c-Ets-1, as noted above, are two known activators bound within the −1350~−1250 region to control *CHOP* induction. These TFs are, moreover, upregulated under ER stress. Therefore, it is reasonable to speculate the presence of a signaling cascade in which AP-1 and c-Ets-1 are engaged in upregulating *ENDOU-1* expression, as an intermediate step, leading to an increase in *CHOP* translation mediated by the increased ENDOU-1 through overriding the inhibitory uORF-trap at 5′-UTR of *CHOP* mRNA, thereby allowing the translation of downstream *CHOP* mRNA during ER stress. Needless to say, such an AP-1/c-Ets-1/*ENDOU-1*/CHOP axis would require further study and confirmation. Apart from the presumed AP-1/c-Ets-1 activators, it is possible that other novel activators might similarly interact with this stress-inducible motif, thereby enabling the control of *ENDOU-1* expression. Further studies to validate these theories could include an electrophoretic mobility shift assay on gel, deletion analysis, mutagenesis experiments, or even yeast one-hybrid screening, as described by [55,56].

Lastly, taking advantage of using bioinformatics could lead to accelerate the discovery of novel promoters and their cognate *cis*-elements of genes. However, how to evaluate variants based on in vitro and in vivo experimental platforms, rule out false positives, and confirm true regulatory motifs is still a major challenge for researchers who are working on promoter analysis. In this study, we used the human *ENDOU-1* gene as an example to demonstrate how to identify *cis*-elements that importantly impact *ENDOU-1* transcription among hundreds of TFBSs predicted by bioinformatics. Our results do contribute to computer algorithms with better prediction of *cis*-elements; they will help to circumvent false-positive elements and efficiently predict cognate transcriptional regulators. Moreover, our studies may contribute to future applications like the design of next-generation of *ENDOU* gene therapies against head and neck squamous cell carcinoma affected by environmental factors [57].

## 3. Materials and Methods

### 3.1. Zebrafish Husbandry and Microscopy

The zebrafish AB strain was cultured as previously described [58]. hpf is defined as normalized hours after fertilization at 28.5 °C, an optimal temperature of culturing. To administer heat-shock treatment to embryos, we followed the protocol described by Lee et al. [28]. Briefly, a 2 mL centrifuge tube filled with 30 dechorionated zebrafish embryos at 72 hpf was subjected to 40 °C for 1 h. Then, the treated embryos were collected into a 3 cm Petri dish and incubated at 28.5 °C until they developed at 96 hpf.

The experiments and treatments of the zebrafish model have been reviewed and approved by the Fu Jen Catholic University and MacKay Medical College Institutional Animal Care and Use Committee with ethics approval numbers A11065 and A1040008, respectively. No specific ethics approval was required for this project as all zebrafish used in this study were between 0 to 5 days post-fertilization. Given the age of the embryos, pain perception has not yet developed at these earlier stages; therefore, this is not considered a painful procedure.

### 3.2. Cell Culture, Transfection, Drug Treatment, and Observation

HEK-293T cells were procured from the Bioresource Collection and Research Center (Taiwan) and were put to use in this study. Cells were cultured and transfected as previously described [59]. For plasmid transfection, we used jetPRIME transfection reagent (Polyplus, Illkirch, France) according to the standard protocol. For drug treatment, cultures were replaced with fresh medium, incubated for 2 h, and the following chemicals individually added: dimethyl sulfoxide (DMSO; D2650, Sigma-Aldrich, St. Louis, MO, USA), which served as the control group, 1 μM Thapsigargin (T9033, Sigma-Aldrich, St. Louis, MO, USA); 20 μM Anisomycin (A9789, Sigma-Aldrich, St. Louis, MO, USA), 150 μM CoCl2 (C2644, Sigma-Aldrich, St. Louis, MO, USA); 250 μM CuSO_4_ (C1297, Sigma-Aldrich, St. Louis, MO, USA); and 100 ng/mL Polyinosinic:polycytidylic acid (Poly I:C; P1530, Sigma-Aldrich, St. Louis, MO, USA), all serving as stress inducers. Cells were then harvested after treatment for 1 to 6 h as indicated.

### 3.3. Dual-Luc Assay

The dual-luc assay performed on cells [59] and zebrafish embryos [60] has previously been described, except a 5 ng/μL of phRG-TK (internal control) and puORF^chop^-luc reporter construct were used. For in vitro studies, we did not assay the luc activity longer than 6 h; this was based on previous studies published by Lee et al. [28], which demonstrated that CHOP protein was first detected after stress treatment for 2 h, while the CHOP protein reached a relatively high level after stress treatment for 6 h. For each experiment in zebrafish, embryos at 96 hpf were collected and divided into three groups to carry out in the vivo Dual-luc assay (n = 60).

### 3.4. Western Blotting

Western blot analysis and immunostaining followed the procedures described previously [28], except that the following antibodies were used: eIF2α (1:1000; CST#5324, Cell Signaling Technology, Danvers, MA, USA); p-eIF2α (1:1000; CST#3398, Cell Signaling Technology, Danvers, MA, USA); zebrafish eIF2α (1:500; GTX124488, Genetex, Irvine, CA, USA); CHOP (1:1000; CST#2895, Cell Signaling Technology, Danvers, MA, USA); and α-tubulin (1∶5000; T9026, Sigma-Aldrich, St. Louis, MO, USA). Fifty μg protein extracts were loaded in each lane to detect the protein level.

### 3.5. Construction of Expression Plasmids

Using specific primers listed in Appendix A, we obtained a 2.1-kb PCR product containing nt −2055~+77 of human *ENDOU-1*, which we then sub-cloned into pGL3-Basic vector (Promega) to generate plasmid pE2.1p. Using restriction enzymes MluI and XhoI, we cut the −2.1~1.2 kb segment from the pE2.1p fragment and thus obtained plasmid pE1.2p, which contains nucleotides −1125~+77 of *ENDOU-1*.

To construct five internal deletion clones, two approaches were used (Appendix A). First, as shown in Appendix A, we employed a two-step PCR strategy to generate four internal deletion constructs fused with luc reporter using restriction enzymes KpnI and StuI (NEB Inc., Ipswich, MA, USA), specific primers (Appendix A), and HiFi polymerase (Roche, Mannheim, Germany), e.g., pE2.1p lacking −1485~−1350 (pE2.1p-d1485/1350). Using the same strategy, we generated pE2.1p-d1850/1750 and pE2.1p-d1749/1650, but we used restriction enzymes KpnI and XhoI (NEB Inc., Ipswich, MA, USA) and corresponding primers (Appendix A). Similarly, we generated pE2.1p-d1350/1250 by using XhoI and StuI and corresponding primers (Appendix A). In a second approach, as shown in Appendix A, we generated plasmid pE2.1p-d1649/1486, and a 0.5 kb PCR product containing an internal MscI cutting site was amplified using primers F5 and R5 (Appendix A). After plasmid pE2.1p was digested by KpnI and MscI (NEB Inc., Ipswich, MA, USA), the resultant linear 6.4 kb DNA fragment was collected and ligated with KpnI-digested 0.4 kb PCR product to generate plasmid pE2.1p-d1649/1486 with 6.8 kb.

### 3.6. Statistics and Reproducibility

All experiments were performed in triplicate, and data were averaged from three independent experiments and presented as mean ± SD. A one-way ANOVA, followed by Tukey’s multiple comparisons test or Student’s *t*-test for comparisons was used to perform statistical analysis. Three levels of significance at * (*p* < 0.05), ** (*p* < 0.005), and *** (*p* < 0.001) were determined.

## Figures and Tables

**Figure 1 ijms-24-17393-f001:**
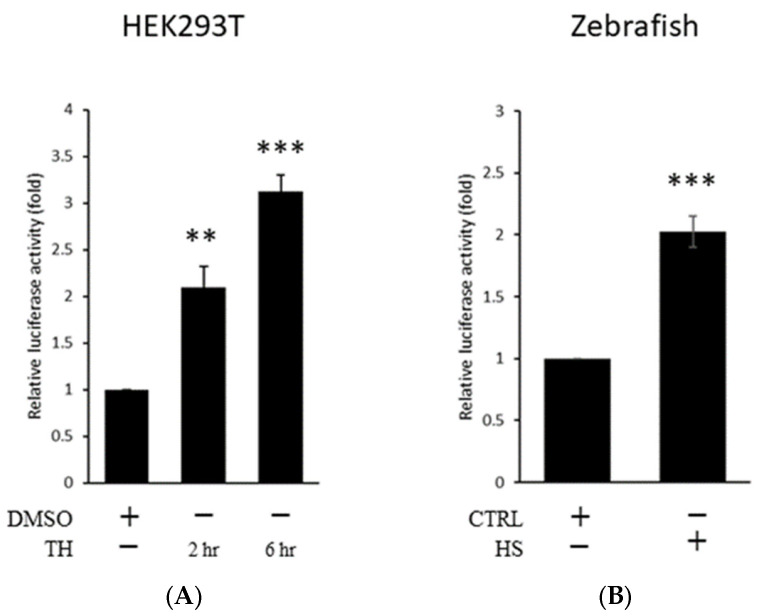
**Effect of the −2055~+77 DNA fragment of human *ENDOU-1* gene on reporter gene expression in stress conditions.** (**A**). Histograms presenting the relative luc activity obtained from HEK-293T cells co-transfected with pE2.1p (100 ng/well) and phRG-TK (20 ng/well) and then treated with either DMSO (control) or Thapsigargin (TH; stress inducer) for 2 or 6 h. HEK-293T cells treated with DMSO served as the control group. The relative luc activity was represented by the fold increase of Flu/Rlu ratio over that obtained from the control group normalized as 1. Values representing DMSO- and TH-treated cells were calculated from four independent experiments and presented as mean ±SD (n = 4). Student’s *t*-test was used to determine significant differences between each group (**, *p* < 0.05; ***, *p* < 0.005). (**B**). Histograms presenting the relative luc activity obtained from zebrafish embryonic cells microinjected at the one-cell stage with pE2.1p (6.9 pg/embryo) combined with phRG-TK (2.3 pg/embryo), followed by heat shock at 72 hpf and luc quantification at 96 hpf. Zebrafish embryos treated with microinjection, but not treated with heat shock, served as the control group. The relative luc activity was represented by the fold increase in the Flu/Rlu ratio over that obtained from the control group normalized as 1. Each value was averaged from three independent experiments and presented as mean ± SD (n = 3). Student’s *t*-test was used to determine significant differences between each group (***, *p* < 0.005).

**Figure 2 ijms-24-17393-f002:**
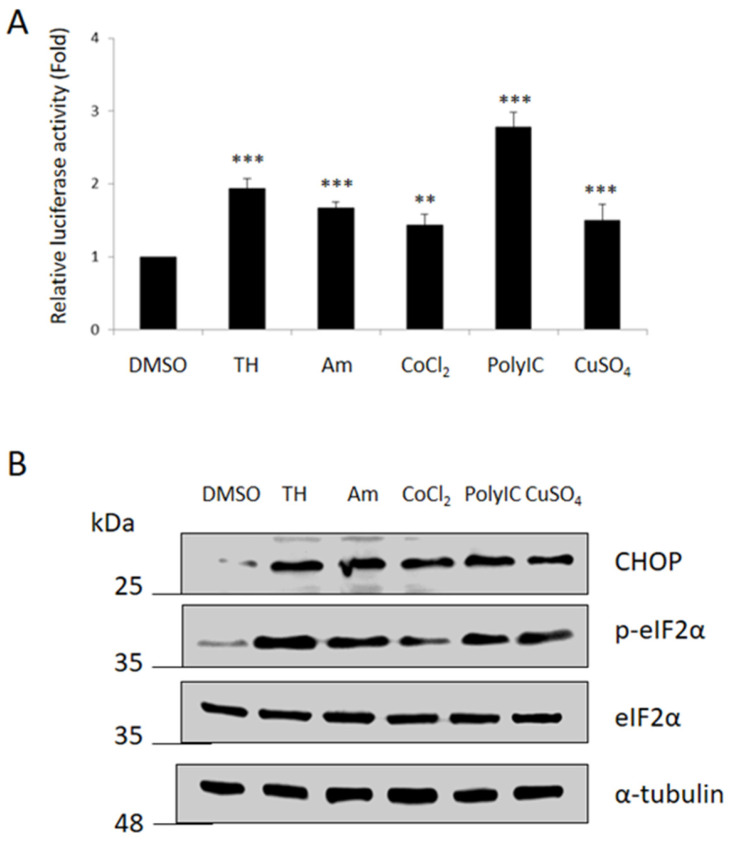
Promoter activity driven by the −2055~+77 DNA fragment of human *ENDOU-1* could be induced by various stress-inducing drugs. (**A**) HEK-293T cells co-transfected with pE2.1p and phRG-TK plasmid for 24 h and then treated with DMSO, Thapsigargin (TH), Anisomycin (Am), CoCl_2_ (Co), Poly I:C (IC), or CuSO_4_ (Cu) for 6 h, followed by analysis of luc activity via a dual-luc assay. Cells treated with DMSO served as a control group. The relative luc activity was represented by the fold increase in the Flu/Rlu ratio over that obtained from the pCS2-transfected control group, normalized as 1. The values were averaged from three independent experiments and presented as mean ± S.D. (n = 3). The Student’s *t*-test was used to perform statistical analysis (***, *p* < 0.001; **, *p* < 0.005) (**B**) Western blot analysis of proteins, as indicated, in HEK-293T cells. The α-tubulin served as internal control.

**Figure 3 ijms-24-17393-f003:**
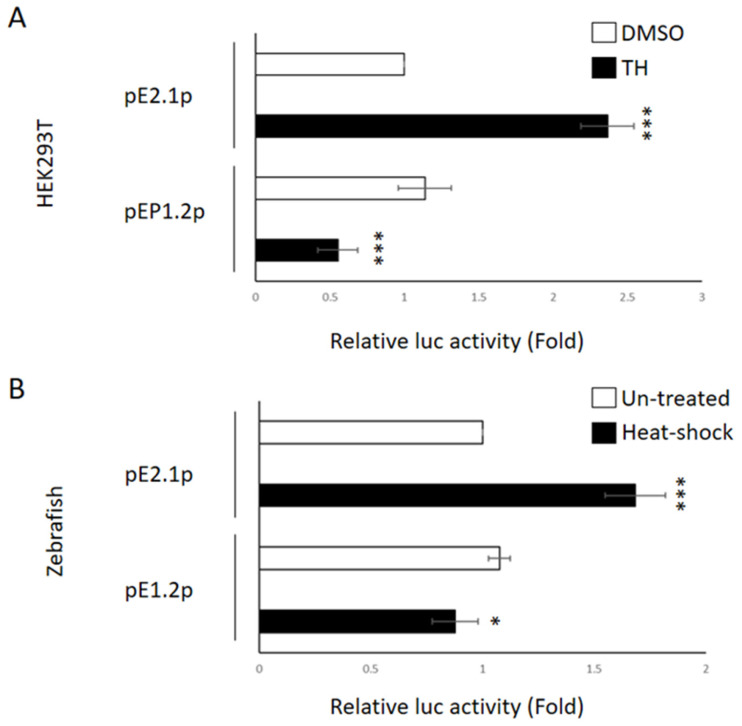
Comparison of luc expression activity driven by the DNA fragments −2055~+77 and −1125~+77 of human *ENDOU-1* gene during ER stress. (**A**) In vitro system. Histograms presenting the relative luc activity obtained from HEK-293T cells cotransfected with either pE2.1p (containing −2055~+77 fragment; 100 ng/well) combined with phRG-TK (20 ng/well) (above) or pE1.2p (containing −1125~+77 fragment; 100 ng/well) combined with phRG-TK (20 ng/well) (bottom) in the absence (empty) or presence (solid) of ER stress inducers. HEK-293T cells treated with DMSO served as the control group. (**B**) In vivo system. Histograms presenting the relative luc activity obtained from zebrafish embryos microinjected at the one-cell stage with either pE2.1p (containing −2055~+77 fragment; 6.9 pg/embryo) combined with phRG-TK (2.3 pg/embryo) (above) or pE1.2p (containing −1125~+77 fragment; 6.9 pg/embryo) combined with phRG-TK (2.3 pg/embryo) (bottom), followed by no treatment (untreated; empty) or heat shock (solid) at 72 hpf and detection of luc activity at 96 hpf. Microinjected zebrafish embryos not exposed to heat shock served as the untreated control group. Relative luc activity was represented by the fold change of Flu/Rlu ratio over that obtained from the control group normalized as 1. Values representing untreated control and heat-shock-treated embryos were calculated from three independent experiments and presented as mean ± SD (n = 3). Student’s *t*-test was used to determine significant differences between pE2.1p- and pE1.2p-injected groups (***, *p* < 0.001; *, *p* < 0.05).

**Figure 4 ijms-24-17393-f004:**
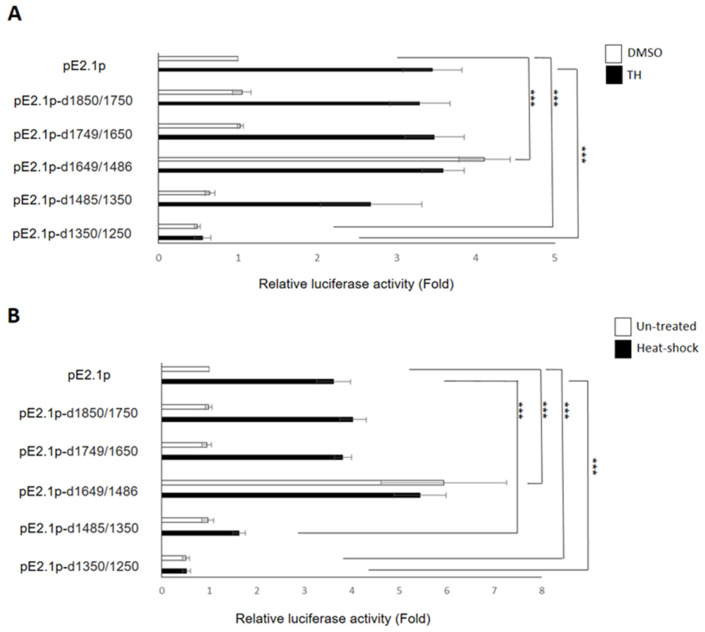
Comparison of the promoter activity of *ENDOU-1* driven by different internal deletion clones from −2.1 kb of the *ENDOU-1* fragment. (**A**) In vitro system. Histograms presented the relative luc activity obtained from HEK-293T cells cotransfected either phRG-TK (20 ng/well) combined with pE2.1p or phRG-TK (20 ng/well) combined with different internal deletions (100 ng/well), as indicated on the left side. For example, plasmid pE2.1p-d1850/1750 contained a −2.1 kb fragment of *ENDOU-1*, but deletion of −1850~−1750 nucleotides. The luc activities obtained from control and five deletion clones, as indicated, were quantified both during normal (DMSO) (represented by empty box) and Thapsigargin (TH)-induced stress condition (represented by solid box). HEK-293T cells treated with DMSO served as control group. (**B**) In vivo system. Histograms presenting the relative luc activity obtained from zebrafish embryos microinjected at the one cell stage with either indicated plasmids (6.9 pg/embryo) and phRG-TK (2.3 pg/embryo) followed by heat-shock treatment at 72 hpf and luminescence detection at 96 hpf. Microinjected zebrafish embryos not exposed to heat shock served as the control group. The relative luc activity was represented by the fold change in the Flu/Rlu ratio over that obtained from the control group normalized as 1. Values representing control and heat-shock-treated embryos were averaged from three independent experiments. A one-way ANOVA, followed by Tukey’s multiple comparison test, was used to perform statistical analysis (***, *p* < 0.001; error bars indicate mean ± SD).

## Data Availability

Data are contained within the article.

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
