# Peer review of "The Upstream 1350~1250 Nucleotide Sequences of the Human ENDOU-1 Gene Contain Critical Cis-Elements Responsible for Upregulating Its Transcription during ER Stress"

_ijms, 2023, doi:10.3390/ijms242417393_

Round 1
Reviewer 1 Report
Comments and Suggestions for Authors
The manuscript titled “The Upstream 1350~1250 Nucleotide Sequences of the Human NEDOU-1 Gene Contain Critical Cis-elements Responsible for Upregulating its Transcription during ER Stress” by Lee, H-C.; et al. is an original scientific work where the authors study the expression of luciferase activities fused on ENDOU-1 genes and deleted upstream segments under stress conditions. The authors found an activator and repressor binding motifs which could have a positive/negative response under these stress conditions, respectively leading the activation or inhibition of ENDOU-1 expression. The study is interesting and it is well-designed.
However, it exists some points that need to be addressed (please, see them below detailed point-by-point). The most relevant outcomes found by the authors can contribute in the growth of many fields like the prognosis of human diseases by the early detection of cellular identification markers. For this reason, I will recommend the present scientific manuscript for further publication in International Journal of Molecular Sciences once all the below described suggestions will be properly fixed.
Here, there exists some points that must be covered in order to improve the scientific quality of the manuscript paper:
1) ABSTRACT. “ENDOU-1 encodes an endoribonuclease (…) 5’-UTR (…)” (lines 12-13). Please, the authors should define the full-name the first time that one term appears. For example, in this case “untranslated region” should be added and then, the abbreviation placed between brackets. This comment should be taken into account for the rest of the main manuscript body text.
2) KEYWORDS. The term “ER stress” is repeated twice. Please, erase one of them. (OPTIONAL) The authors should consider to add “HEK 293T cells” corresponding to the in vitro experiments in the keyword list.
3) INTRODUCTION. “The ISR is primarily (…) adaptative cellular response to stress” (lines 39-41). Here, it is neccesary to furnish some relevant example where the oxidative stress leads to DNA degradation [1, 2] and how living cells through ISR strategies attempt to overcome this situation.
[1] Ryter, S.W.; Kim, H.P.; Hoetzel, A.; Park, J.W.; Nakahira, K.; Wang, X.; Choi, A.M.K. Mechanisms of Cell Death in Oxidative Stress. Antioxid. Redox. Signal. 2007, 9, 49-89. https://doi.org/10.1089/ars.2007.9.49.
[2] Novo, N.; Romero-Tamayo, S.; Marcuello, C.; Boneta, S.; Blasco-Machín, I.; Velázquez-Campoy, A.; Villanueva, R.; Moreno-Loshuertos, R.; Lostao, A.; Medina, M.; et al. Beyond a platform protein for the degradosome assembly: The Apoptosis-Inducing Factor as an efficient nuclease involved in chromatinolysis. PNAS Nexus 2023, 2, pgac312. https://doi.org/10.1093/pnasnexus/pgac312.
4) “Dysregulated ISR signaling (…) cognitive disorders, neurodegenerative disorders (…) metabolic disorders” (lines 42-43). Please, the authors should try to avoid repetitions. In this context, it may be desirable to change some “disorder” terms by anothers like “malignancies” or “diseases”.
5) RESULTS & DISCUSSION. “The dual-luc ativity was performed (…) for 2 or 6 hr, followed by analysis of luc activity” (lines 95-98). Why did the authors not assay the luciferase actifity for longer times? A brief statement should be provided in this regard.
6) “Zebrafish embryos (…) heat shock at 72 hpf” (lines 103-105). Please, the authors should define the term “hours post fertilization”.
7) Figure 1 (line 111). Why did the negative control measurements conducted at the initial time not show the standard deviation (SD) bards? Then, it would be desirable to homogenize the Y-axis values to better compare the luciferase activity between both examined conditions shown in panels A) and B). Same comment for the Fig. 2 A) (line 157).
8) MATERIALS & METHODS. The authors should define the full-details (name and country) of all the supplier companies for all the consumables, chemical reagents and techniques used in this research.
9) CONCLUSIONS (OPTIONAL). The authors should consider to add a short section to highlight the most relevant outcomes found in this work and some potential future applications like the design of the next-generation of gene therapies against human diseases affected by the environmental factors [3].
[3] Xu, C.; Zhang, Y.; Shen, Y.; Shi, Y.; Zhang, M.; Zhou, L. Integrated Analysis Reveals ENDOU as a Biomarker in Head and Neck Squamous Cell Carcinoma Progression. Front. Oncol. 2021, 10, 522332. https://doi.org/10.3389/fonc.2020.522332.
10) REFERENCES. The references are mostly in the proper format style of International Journal of Molecular Sciences. The publication year should be highlighted in bold.
Comments on the Quality of English Language
The manuscript is well-written. However, it may be desirable a final check by all the authors.
Author Response
1) “ENDOU-1 encodes an endoribonuclease (…) 5’-UTR (…)” (lines 12-13). Please, the authors should define the full-name the first time that one term appears. For example, in this case “untranslated region” should be added and then, the abbreviation placed between brackets. This comment should be taken into account for the rest of the main manuscript body text.
Ans: Thank you for your suggestion. We corrected these mistakes. Please see lines 12 to 13 of page 1.
2) The term “ER stress” is repeated twice. Please, erase one of them. (OPTIONAL) The authors should consider to add “HEK 293T cells” corresponding to the in vitroexperiments in the keyword list.
Ans: Thank you for pointing out this mistake. We corrected it. Please see line 39 of page 1. Additionally, as your suggestion, we added HEK-293T cells in the keyword list.
3) “The ISR is primarily (…) adaptative cellular response to stress” (lines 39-41). Here, it is neccesary to furnish some relevant example where the oxidative stress leads to DNA degradation [1, 2] and how living cells through ISR strategies attempt to overcome this situation. [1] Ryter, S.W.; Kim, H.P.; Hoetzel, A.; Park, J.W.; Nakahira, K.; Wang, X.; Choi, A.M.K. Mechanisms of Cell Death in Oxidative Stress. Antioxid. Redox. Signal. 2007, 9, 49-89. https://doi.org/10.1089/ars.2007.9.49. [2] Novo, N.; Romero-Tamayo, S.; Marcuello, C.; Boneta, S.; Blasco-Machín, I.; Velázquez-Campoy, A.; Villanueva, R.; Moreno-Loshuertos, R.; Lostao, A.; Medina, M.; et al. Beyond a platform protein for the degradosome assembly: The Apoptosis-Inducing Factor as an efficient nuclease involved in chromatinolysis. PNAS Nexus2023, 2, pgac312. https://doi.org/10.1093/pnasnexus/pgac312.
Ans: As your suggestion, we added these two references in the revised manuscript. Meanwhile, we also added some statements in the main text as follows: The ISR is primarily a pro-survival homeostatic program, aiming to optimize adaptive cellular response to stress. The ISR restores cellular balance by reprogramming gene expressions in response to different environmental and pathological conditions, including oxidative stress, proteostasis, viral infection and nutrient deprivation [13, 14]. However, once the stress cannot be removed, the ISR drives signaling toward cell death, such as the Apoptosis-Inducing Factors which will act as a prodeath effector to trigger DNA cleavage and parthanatos [15]. For example, dysregulated ISR signaling can be involved in cognitive disturbances, neurodegenerative diseases, tumor malignancies, diabetes and metabolic imbalances [16]. Please see line 40 of page 1 to line 48 of page 2.
4) “Dysregulated ISR signaling (…) cognitive disorders, neurodegenerative disorders (…) metabolic disorders” (lines 42-43). Please, the authors should try to avoid repetitions. In this context, it may be desirable to change some “disorder” terms by anothers like “malignancies” or “diseases”.
Ans: In response to your suggestion, we rewrote this sentence as follow: “For example, dysregulated ISR signaling can be involved in cognitive disturbances, neurodegenerative diseases, tumor malignancies, diabetes and metabolic imbalances.” Please see lines 47 to 48 of page 2.
5) RESULTS & DISCUSSION. “The dual-luc ativity was performed (…) for 2 or 6 hr, followed by analysis of luc activity” (lines 95-98). Why did the authors not assay the luciferase actifity for longer times? A brief statement should be provided in this regard.
Ans: The reason why we did not assay the luciferase activity longer than 6 hr was that based on our previous studies published by Lee et al. (2021) which demonstrated that CHOP protein was first detected after stress treatment for 2 hr, while the CHOP protein reached to a relatively high level after stress treatment for 6 hr (please see figure below which was Figure EV2C in EMBO J). Moreover, comparing the expressions between 2-hr stress (weak) and 6-hr stress (strong) should provide insight for us to know whether the promoter activity is positively correlated to stress duration. Therefore, we did not assay the luciferase activity longer than 6-hr stress treatment. We added these statements in the Materials and Methods. Please see lines 414 to 417 of page 11.
(Lee et al., 2021, Figure EV2C)
6) “Zebrafish embryos (…) heat shock at 72 hpf” (lines 103-105). Please, the authors should define the term “hours post fertilization”.
Ans: Thank you. The “hours post fertilization” (hpf) is defined as normalized hours after fertilization at 28.5℃, an optimal temperature of culturing. We added this brief description of hpf in the Materials and Methods. Please see line 111 of page3 and lines 385 to 386 of page 10.
hpf is defined as normalized hours after fertilization at 28.5℃, an optimal temperature of culturing
7) (A) Figure 1 (line 111). Why did the negative control measurements conducted at the initial time not show the standard deviation (SD) bards? (B)Then, it would be desirable to homogenize the Y-axis values to better compare the luciferase activity between both examined conditions shown in panels A) and B). Same comment for the Fig. 2 A) (line 157).
Ans: (A) Because the luciferase activity of the negative control group was normalized as 1 each time in order to compare with the luciferase activities obtained from the other two experimental groups.
(B) Y-axis values are the relative luciferase activities represented by the fold increase of Fluc/Rluc ratio over that obtained from the control group which was normalized as 1. For example, As the raw data presented in Figure 1A (please see figure below), the original Fluc/Rluc values of each negative control group were 32.76218, 38.05428 and 33.87533. The original Flu/Rlu values of each TH-2hr-treatment group are 76.20837, 69.44760 and 68.17035, respectively.
After we normalized the luciferase activity obtained from each negative control group as 1, followed by making a comparison with the value from the TH-2hr-treated group. The result showed that the luciferase activity of the TH-2hr-treated group was an average of 2.05-fold higher than that of the control group (please see figure below).
8) MATERIALS & METHODS. The authors should define the full-details (name and country) of all the supplier companies for all the consumables, chemical reagents and techniques used in this research.
Ans: Thank you. We added all required information in the revised manuscript. Please see the revised Materials and Methods section.
9) CONCLUSIONS (OPTIONAL). The authors should consider to add a short section to highlight the most relevant outcomes found in this work and some potential future applications like the design of the next-generation of gene therapies against human diseases affected by the environmental factors [3].[3] Xu, C.; Zhang, Y.; Shen, Y.; Shi, Y.; Zhang, M.; Zhou, L. Integrated Analysis Reveals ENDOU as a Biomarker in Head and Neck Squamous Cell Carcinoma Progression. Front. Oncol. 2021, 10, 522332. https://doi.org/10.3389/fonc.2020.522332.
Ans: In response of your suggestion, we highlighted the most relevant outcomes found in this work and added some potential future applications in the Conclusion of revised manuscript as follows: “Lastly, taking advantage of using bioinformatics could lead to accelerate the dis-covery of novel promoters and their cognate cis-elements of genes. However, how to evaluate variants based on in vitro and in vivo experimental platforms, rule out false positive, and confirm true regulatory motifs are still a major challenge for researchers who are working on promoter analysis. In this study, we used human ENDOU-1 gene as an example to demonstrate how to identify cis-elements that importantly impact on the ENDOU-1 transcription among hundreds of binding sites predicted by bioinfor-matics. Our studies do contribute to computer algorithms with better prediction of cis-elements, circumvent the false positive elements, and efficiently predict cognate transcriptional regulators. Moreover, our studies may contribute future applications like the design of the next-generation of ENDOU gene therapies against head and neck squamous cell carcinoma affected by the environmental factors [57]. Please see lines 371 to 382 of page 10.
10) REFERENCES. The references are mostly in the proper format style of International Journal of Molecular Sciences. The publication year should be highlighted in bold.
ANS: Thank you. We corrected them.
Reviewer 2 Report
Comments and Suggestions for Authors
The manuscript by Lee et al. investigates the transcriptional control of ENDOU-1, a gene encoding an endoribonuclease crucial for overcoming the inhibitory uORF-trap in the 5’ UTR of CHOP transcript during ER stress. The authors cloned a 2.1 kb upstream region of human ENDOU-1 and found that this promoter (pE2.1p) significantly upregulates transcription under stress conditions in both transfected human cells and zebrafish embryos. Further analysis suggests that cis-elements within the -2055~-1125 segment of the promoter play a critical role in ENDOU-1 upregulation, with a stress-inducible activator likely binding to the -1350~-1250 segment. The manuscript is well described, and I recommend it for publication after minor changes.
1. The resolution of Figure S1 is not clear, I guess the author may represent the list of all putative TF-binding in table format.
2. Figure 5 should be shifted to supplementary figures.
Author Response
1) The resolution of Figure S1 is not clear, I guess the author may represent the list of all putative TF-binding in table format.
ANS: In response to your suggestion, we used the table format to list all putative TF-binding instead of the figure format. Please see the revised Table S1
2) Figure 5 should be shifted to supplementary figures.
ANS: In response to your suggestion, we shifted Figure 5 to Supplementary figure S2.